# Effects of Methoxyfenozide-Loaded Fluorescent Mesoporous Silica Nanoparticles on *Plutella xylostella* (L.) (Lepidoptera: Plutellidae) Mortality and Detoxification Enzyme Levels Activities

**DOI:** 10.3390/ijms23105790

**Published:** 2022-05-21

**Authors:** Muhammad Bilal, Muhammad Umair Sial, Lidong Cao, Qiliang Huang

**Affiliations:** 1Institute of Plant Protection, Chinese Academy of Agricultural Sciences, Beijing 100193, China; muhammadentomologist@gmail.com (M.B.); caolidong@caas.cn (L.C.); 2Department of Entomology, University of Agriculture, Faisalabad 38000, Pakistan; omairsial@hotmail.com

**Keywords:** *Plutella xylostella*, nanotechnology, fluorescent silica nanoparticles, detoxification enzymes, insecticide resistance

## Abstract

The diamond back moth, *Plutella xylostella*, causes severe damage at all crop stages, beside its rising resistance to all insecticides. The objective of this study was to look for a new control strategy such as application of insecticide-loaded carbon dot-embedded fluorescent mesoporous silica nanoparticles (FL-SiO_2_ NPs). Two different-sized methoxyfenozide-loaded nanoparticles (Me@FL-SiO_2_ NPs-70 nm, Me@FL-SiO_2_ NPs-150 nm) were prepared, with loading content 15% and 16%. Methoxyfenozide was released constantly from Me@FL-SiO_2_ NPs only at specific optimum pH 7.5. The release of methoxyfenozide from Me@FL-SiO_2_ NPs was not observed other than this optimum pH, and therefore, we checked and controlled a single release condition to look out for the different particle sizes of insecticide-loaded NPs. This pH-responsive release pattern can find potential application in sustainable plant protection. Moreover, the lethal concentration of the LC_50_ value was 24 mg/L for methoxyfenozide (TC), 14 mg/L for Me@FL-SiO_2_ NPs-70 nm, and 15 mg/L for Me@FL-SiO_2_ NPs-150 nm after 72 h exposure, respectively. After calculating the LC_50_, the results predicted that Me@FL-SiO_2_ NPs-70 nm and Me@FL-SiO_2_ NPs-150 nm exhibited better insecticidal activity against *P. xylostella* than methoxyfenozide under the same concentrations of active ingredient applied. Moreover, the activities of detoxification enzymes of *P. xylostella* were suppressed by treatment with insecticide-loaded NPs, which showed that NPs could also be involved in reduction of enzymes. Furthermore, the entering of FL-SiO_2_ NPs into the midgut of *P. xylostella* was confirmed by confocal laser scanning microscope (CLSM). For comparison, *P. xylostella* under treatment with water as control was also observed under CLSM. The control exhibited no fluorescent signal, while the larvae treated with FL-SiO_2_ NPs showed strong fluorescence under a laser excitation wavelength of 448 nm. The reduced enzyme activities as well as higher cuticular penetration in insects indicate that the nano-based delivery system of insecticide could be potentially applied in insecticide resistance management.

## 1. Introduction

The diamondback moth, *Plutella xylostella* (L.) (Lepidoptera: Plutellidae), is a significant and cosmopolitan pest of many cruciferous vegetables [1]. The annual control and damage costs for this insect pest were expected at USD 4–5 billion worldwide [1]. Presently, chemical control remains the most important and widely used strategy against *P. xylostella* [1,2], now considered a well-known pest throughout the world due to the development of resistance against all types of insecticides, organic and synthetic, e.g., pyrethroids, organophosphates, carbamates [3,4]. Different insecticides are applied to repel, destroy, or decrease pest infestations to save crops from insect pests and increase the yield [5]. According to various statistical data, about 30% of total worldwide crop production is saved every year by using different insecticides [5]. Moreover, the improper disposal and extensive use of insecticides affect several possible sources in air, food, and water, causing accumulation of toxins in the environment and poisonous threats to humans and animals [5,6,7]. Insecticide residuals not only affect human health but also harm biodiversity as well as ecological surroundings [8,9,10,11,12].

When insecticides are applied to crops, the foliage is the first contact part and insecticide is then transferred to other plant parts [13]. Previously, it was reported that less than 10% of insecticides are utilized by the crop leaf, while the rest runs off into the nearby surroundings, and about 0.1% of an applied insecticide reaches the target insect [13,14]. Therefore, it is very important to reduce losses by increasing adhesion and deposition of insecticides into the plant foliage. An insecticide formulation with strong deposition and adhesion to the plant foliar could be required for increasing utilization on the plant surface and producing the successful application rate [15,16].

The role of nanotechnology for crop protection nano-based insecticides has attracted agriculture scientists for significant control and management of devastating polyphagous insects in agriculture [17]. In current years, use of nanotechnological systems has been developed for agricultural applications, operating intensive research and development practices at both industrial and academic levels [18,19]. Innovations in nanotechnology can help to alleviate the complications related to the application technologies of classical insecticide formulations [18]. Moreover, insecticide-loaded nanoparticles decrease the dose of insecticides, reduce nutrient losses, identify plant-damaging insects and pathogens, detect insecticide residue, and increase plant yields without harming the environment and non-target beneficial organisms [17]. Due to their small size and large surface-area-to-volume ratio, nano-insecticides display exceptional characteristics compared to their bulk counterparts [20,21].

Fortunately, recent studies have been working on nanotechnology to produce residue-free nano-formulations of insecticides with minimum environmental persistence and better insecticidal activity as well as the least negative effect to the environment and human health [22,23]. In addition, the application of nano-based insecticides can delay the development of resistance against the insecticides [24]. Recently, various nano-formulations with different nanoparticles have been developed, specifically, silica MgO, CuO, Ag, and ZnO nanoparticles with confirmed better insecticidal activity [25].

In recent research, the nano-formulation of insecticide has proven the higher efficiency for insect control with lesser harmful effects on the environment as compared with technical insecticide [26]. Methoxyfenozide is a diacylhydrazine insecticide, highly effective against lepidopteran larvae, chemically safe for environment and is used to control many insects [27,28]. The active molecules can potentially mimic the action of two insect hormones, the sesquiterpenoid juvenile hormone (JH) and the steroidal molting hormones, 20-hydroxyecdysone (20E), which regulate different stages of development and growth of insects [29]. Furthermore, they can mimic the action of molting hormones and induce an incomplete and precocious molt in many insects through the action of ecdysteroid receptor proteins [30]. Diacyl hydrazine such as methoxyfenozide shows green compounds having characters to control target pests and have almost no impact on most beneficial and non-target organisms [30]. Currently, out of five registered diacyl hydrazine insecticides, namely, Fufenozide (JS-118), Tebufenozide (RH-5992), Chromafenozide (ANS-118, CM-001), methoxyfenozide (RH-2485), and halofenozide (RH-0345). Methoxyfenozide is the most widely registered diacyl hydrazine class of insecticide, with registrations in more than fifty countries for control of insects on several crops [17].

For these reasons, methoxyfenozide was selected as a model insecticide to prepare the controlled release formulations (CRFs) using fluorescent MSNs (FL-SiO_2_ NPs) as nanocarriers. The strong and stable luminescence of the as-prepared FL-SiO_2_ NPs originates from the carbon dots generated from the calcination [31]. The methoxyfenozide-loaded FL-SiO_2_ nanoparticles (Me@FL-SiO_2_ NPs) were fully characterized including morphological and mesoporous features, thermal stability, in vitro release patterns, and insecticidal activity against *P. xylostella*. More importantly, the effects of Me@FL-SiO_2_ NPs on the detoxification enzyme activities, including glutathione S-transferase (GST), carboxylesterase (CarE), acetylcholinesterase (AChE), and cytochrome P450 monooxygenase (P450), were studied. Moreover, the insecticidal activity of Me@FL-SiO_2_ NPs against *P. xylostella* was visually observed with confocal laser scanning microscopy (CLSM). The present research seeks to explore the feasibility of nanocarrier platforms for overcoming insecticide resistance and potential application in sustainable plant protection.

## 2. Materials and Methods

### 2.1. Reagents and Chemicals

Tetraethoxysilane (TEOS, 99%) was purchased from Fluorochem Ltd. (Hadfield, UK), while N-[3-(trimethoxysily) propyl] ethylenediamine (PEDA-TMS) and cetyltrimethylammonium bromide (CTAB 99%) were bought from J&K Scientific Ltd. (Beijing, China). The model insecticide, methoxyfenozide technical concentrate (TC) (95%), was supplied by Taizhou Dapeng Pharmaceutical Co., Ltd. (Taizhou, China). *Plutella xylostella* larvae were taken from the Insect Culture Lab in the Institute of Plant Protection, Chinese Academy of Agricultural Sciences (Beijing, China). All other chemicals and reagents were commercially provided and used as obtained. Milli-Q water distillation system (Millipore Corporation, Bedford, MA, USA) provided deionized water.

### 2.2. Preparation of Fluorescent Silica Nanoparticles (FL-SiO_2_ NPs)

Two types of fluorescent silica nanoparticles (FL-SiO_2_ NPs) with 70 nm and 150 nm sizes were prepared following the procedure reported by Bilal [32]. Cetyltrimethylammonium bromide was dissolved in deionized water under constant magnetic stirring with 1000 rpm, followed by the addition of sodium hydroxide (NaOH) solution. The resultant solution was heated up to 80 °C in an oil bath. Thereafter, TEOS, ethylacetate, and PEDA-TMS were added dropwise and the mixture was stirred at 80 °C for 2 h. The raw product was centrifuged and washed three times with ethanol. To remove CTAB, 80 µL of concentrated HCl was added and the solution was stirred at 60 °C for 6 h. The as-synthesized white product was again washed twice with ethanol, and dried at 80 °C overnight. To obtain fluorophore-free luminescent MSNs, the powders were calcined in a muffle furnace at 400 °C for 2 h in open air.

### 2.3. Preparation of Insecticide-Loaded Nanoparticles HCl

Methoxyfenozide was loaded into FL-SiO_2_ NPs according to the previously described procedures [32,33]. Normally, about 1.2 g of methoxyfenozide was dissolved in 40 mL of methanol (CH_3_OH) to obtain a methoxyfenozide solution with a 30 mg/mL concentration. FL-SiO_2_ NPs with different carrier/insecticide ratios were subsequently dispersed in the above-mentioned solution. The suspension was magnetically stirred for 6 h at room temperature; thereafter, methoxyfenozide-loaded FL-SiO_2_ NPs (Me@FL-SiO_2_ NPs) were obtained by filtration. Finally, the as-prepared Me@FL-SiO_2_ NPs were dried overnight at 60 °C to remove the solvent.

### 2.4. Sample Characterization

The morphological and structural characteristics of FL-SiO_2_ NPs were determined by transmission and scanning electron microscopy (TEM, Tecnai G2, F20 S-TWIN, FEI, Hillsboro, OR, USA, accelerated at 200 kV and SEM, SU8010, Hitachi Ltd., Tokyo, Japan, operated at 10 kV voltages).

Spectrophotometer NICOLET 6700 (Thermo Scientific, Waltham, MA, USA) was used to analyze the Fourier transform infrared (FTIR) of methoxyfenozide, FL-SiO_2_ NPs, and Me@FL-SiO_2_ NPs using potassium bromide pellets within spectral regions of 4000–400 cm^−1^.

Thermogravimetric analysis (TGA) was performed on a Perkin Elmer Pyris Diamond thermogravimetric analyzer (Woodland, CA, USA) within a 20 °C to 550 °C temperature range in the presence of a N_2_ atmosphere.

The particular characteristics such as pore size and surface areas of the samples were measured by the nitrogen adsorption using pore size and a specific surface area analyzer (TriStarII 3020, Micromeritics Instruments Crop, Norcross, GA, USA) at −196 °C. The specific surface area was determined using equation of Brunauer–Emmett–Teller (BET) [34], while the pore size was calculated from the Barrett–Joyner–Halenda (BJH) [35] model using the adsorption data of isotherms.

### 2.5. Methoxyfenozide Loading Content 

Methoxyfenozide loading content of the nanoparticles was measured by high-performance liquid chromatography (HPLC, 1200-DAD (Diode Array Detector), Agilent Santa Clara, CA, USA) using the method reported by Bilal et al. (2020) [32]. Normally, loading content was determined through HPLC analysis after vigorous mixing of the 10 mg of Me@FL-SiO_2_ NPs in 10 mL of acetonitrile. The HPLC operating operations were as follows: mobile phase (acetonitrile/0.1% formic acid (*v*/*v*) = 80:20), reversed phase column-C_18_ (4.65 m × 250 mm × 5 μm, Bonna-Agela Technologies Inc., Tianjin, China); Column temperature, 30 °C; flow rate, 1 mL/min, and detection wavelength, 275 nm. The loading content (%) and encapsulation efficacy (%) of methoxyfenozide were evaluated as: loading content (%) = (weight of methoxyfenozide encapsulated in nanoparticles/weight of nanoparticles) × 100; encapsulation efficacy (%) = (weight of methoxyfenozide in nanoparticles/initial weight of methoxyfenozide used) × 100.

### 2.6. In Vitro Release Experiment

The release behavior of methoxyfenozide was studied from the prepared Me@FL-SiO_2_ NPs-70 nm and Me@FL-SiO_2_ NPs-150 nm at pH 7.5, comprising phosphate buffer saline (PBS), acetonitrile, and Tween-80 (70:29.5:0.5, *v*/*v*/*v*). About 20 mg of each size of Me@FL-SiO_2_ NPs was put in the dialysis bag (molecular weight cut-off, 8000 Da), which was placed in 200 mL of release medium in a dissolution tester (D-800LS, Tianjin University, Tianjin, China) at 25 ± 1 °C with a stirring speed of 150 rpm. The release of methoxyfenozide was observed by measuring the concentration of methoxyfenozide in the release solution at different time intervals. At different time intervals, 0.8 mL of the release medium was taken out for HPLC analysis, and the same volume of fresh release solution was added. The release of methoxyfenozide was calculated using the following equation:$$E_{r}=\frac{V_{e}\sum_{i=0}^{n-1}Ci+V_{0}C_{n}}{m_{insecticide}}\times 100\%$$
where *E_r_* shows the release (%) of methoxyfenozide with respect to the loaded insecticide; *C_n_* is the methoxyfenozide concentration (mg/mL) in the release medium at time *n*; *V_e_* (0.8 mL) is the sample volume taken at various time intervals; *m_insecticide_* is the total quantity (mg) of insecticide carrying with the NPs; and *V*_0_ (200 mL) is the volume of release solution.

### 2.7. Insecticidal Activity of Me@FL-SiO_2_-70 nm and -150 nm NPs against P. xylostella 

The in vivo toxicological effect of both sizes of Me@FL-SiO_2_ NPs against *P. xylostella* was checked with bioassay using the leaf dip method as described previously, with slight changes [32]. *Plutella xylostella* larvae were reared on wild cabbage, *Brassica oleracea* (capitata), at temperature of 25 ± 1 °C, relative humidity of 70 ± 10%, with photoperiod (16L: 8D). After the preliminary experiment, five various concentrations of methoxyfenozide, Me@FL-SiO_2_-70 nm, and Me@FL-SiO_2_-150 nm, were set as 200, 100, 50, 25, and 12.5 mg/L for each treatment. After dipping the leaves for 20 s in different methoxyfenozide concentrations, they were dried at room temperature for 2 h. Filter paper was placed in each petri dish with a diameter of 10 cm to prevent the leaves from dehydrating. Treated leaves were placed in the bottom of prepared petri dishes. Each treatment was performed with 4 replicates. The mortality data were taken after 72 h of the bioassay. For a control group, deionized water and methoxyfenozide (TC) were also used under the same method for bioactivity testing.

### 2.8. Confocal Study of FL-SiO_2_ NPs in the Midgut of P. xylostella 

Third-instar live insect samples, after treating with blank carrier material of FL-SiO_2_ NPs, were taken for fluorescence imaging 72 h after exposure. The previously described dissection method [32,36,37] was used to remove the midgut of larvae for visual observation of fluorescent NPs in the *P. xylostella* body. The larvae were surface-sterilized for 5 s in 95% ethanol before dissection. The midgut was carefully removed from the cuticle after cutting the larval body laterally with the help of dissecting forceps. The gut was carefully washed with PBS solution (pH 7.4), slightly placed on a 3 mm glass slide, covered with a cover slip, and then the FL-SiO_2_ NPs was observed under CLSM (TCS SP8, Leica, Germany) with a laser excitation wavelength of 448 nm.

### 2.9. Detoxification Enzymes

#### 2.9.1. Preparation of *P. xylostella* Samples

The detoxification enzyme activities in *P. xylostella* samples treated with methoxyfenozide TC and Me@FL-SiO_2_ NPs were observed in a Corning^®^ 96 Well Clear Polystyrene Microplate, after preparing the homogenates following the methodology [38,39]. About five third-instar of *P. xylostella* larval populations were crushed with mortar pestle and homogenized on ice in 1.5 mL of 0.1 M PBS (pH 7.8), comprising 1 mM phenyl thio urea (PTU), 1 mM phenylmethyl sulfonyl fluoride (PMSF), 1 mM dithiothreitol (DTT), and 1 mM ethylene diamide tetra acetic acid (EDTA). After the centrifugation of homogenates at 12,000 rpm for 15 min, the supernatants were collected at 4 °C. The resulting supernatant was stored at −20 °C and used as an enzyme source. The Bradford method assay was used to determine the total protein in the samples, using the bovine serum albumin (BSA) as a standard [40].

#### 2.9.2. Activity of Glutathione-S-Transferase (GST)

The method reported by Kristensen was followed to determine the GST activity using 30 mM of CDNB (1-chloro-2, 4-dinitrobenzene) as a substrate [41]. The enzyme solution was prepared by using PBS (66 mM, pH 7.4) and 50 mM reduced glutathione (GSH). GST activity was noted using a BioTek^®^ microplate reader at a wavelength of 340 nm at 27 °C for 5 min with 30 s intervals. The GST values were displayed as ΔmOD_340_ min^−1^ mg protein^−1^.

#### 2.9.3. Activity of Carboxylesterase (CarE)

CarE activity was measured using α-naphthyl acetate as a substrate using the method of Shen et al. (2020) [42]. About 0.3 mM α-naphthyl acetate and 10^−3^ mM physostigmine (an inhibitor of acetylcholinesterase) were used for preparation of the substrate. Prior to addition, the enzyme source was diluted 20 times in 1.0 mL of PBS (0.04 M, pH 7.0) and the resultant mixture was incubated at 30 °C with slight shaking for half an hour. The reaction mixture was stopped by addition of 1.0 mL of distilled water comprising fast blue B salt and sodium dodecyl sulphate. A BioTek^®^ microplate reader was used for measuring the changes in absorbance at 600 nm after 30 min. The CarE activity results were displayed as ΔmOD_600_ min^−1^ mg protein^−1^.

#### 2.9.4. P450 Activity

The P450 activity levels in the samples were determined using an insect mixed-function oxidase (MFO) ELISA kit (Huabaitai Biotechnology Corporation, Beijing, China), following the method used by Cui [43]. According to the instructions of the manufacturer, the enzyme sources were moved to the microcells of the ELISA kit. The ODs of treated samples and standard were measured by Microplate reader (Synergy HT multi-mode) at a wavelength of 450 nm. At first, the standard curve was constructed by plotting the OD values against the concentrations of the working standard solutions. The P450 levels were described as ΔmOD_450_ min^−1^ mg protein^−1^.

#### 2.9.5. AChE Activity

AChE activity was determined using acetylthiocholine iodide as the substrate, following a previous procedure [44,45]. Enzyme solution was prepared by using chemicals, i.e., 0.1 M DTNB (5,5-dithiobis-2-nitrobenzoic acid), 75 mM acetylthiocholine iodide, and phosphate-buffered saline (0.1 M, pH 7.4). The prepared solution was incubated at 27 °C for 15 min after slight shaking. About 1 mM physostigmine was used to stop the reaction mixture and changes in absorbance were measured at 412 nm using a BioTek^®^ microplate reader. The AChE activity results were expressed as ΔmOD_412_ min^−1^ mg protein^−1^.

#### 2.9.6. Statistical Analysis

The mortalities and enzymatic levels were calculated using Microsoft Excel 2010. The lethal concentration values (LC_50_) and (LC_90_) and their confidence limits (CL) were calculated using IBM SPSS software. Enzyme analysis was also performed through IBM SPSS software. ZetaSizer Nano ZS Analyzer, based on DLS, was used to measure the diameters of Me@FL-SiO_2_ NPs and FL-SiO_2_ NPs. Using the statistical analysis of SEM micrographs of more than 200 nanoparticles, the average diameters of FL-SiO_2_ NPs-70 nm and FL-SiO_2_ NPs-150 nm were calculated, while Origin Pro-8 software was used for plotting graphs, and differences between the treatments were calculated by Tukey’s test (*p* ≤ 0.05).

## 3. Results

### 3.1. Morphological and Structural Characterization

The potential controlled release characteristics of silica nanoparticles have received considerable attentions because of their application in smart delivery systems [46,47]. Recently, two differently sized FL-SiO_2_ NPs, FL-SiO_2_ NPs-0 nm and FL-SiO_2_ NPs-150 nm, were prepared using surfactant CTAB as a structure-directing agent, with TEOS used as a silica precursor. The carbon dots formed during the calcination process in preparation of FL-SiO_2_ NPs produce the stable and strong fluorescence which is essential for in vivo imagining of nanoparticles as delivery vehicles in biological systems [48,49,50]. Methoxyfenozide was loaded into FL-SiO_2_ NPs by simple immersion in a CH_3_OH solution by magnetically stirring at room temperature. In the optimum circumstances of carrier/insecticide ratio 2:1, the loading content of methoxyfenozide was calculated up to 15% and 16%. The structural and morphological characters of the two sizes of FL-SiO_2_ NPs and Me@FL-SiO_2_ NPs were checked under TEM and SEM (Figure 1). 

The SEM images exhibited smooth surfaces with monodispersed spherical structure of FL-SiO_2_ NPs [51,52]. There were no specific variances between the Me@FL-SiO_2_ NPs and FL-SiO_2_ NPs with respect to particle size and surface roughness. Using the statistical analysis of SEM micrographs of more than 200 nanoparticles, the average diameters of FL-SiO_2_ NPs-70 nm and FL-SiO_2_ NPs-150 nm were 102 and 168 nm, while 113 nm and 180 nm sizes were determined in the case of Me@FL-SiO_2_ NPs-70 nm and Me@FL-SiO_2_ NPs-150 nm. Mesoporous nature and morphological features of FL-SiO_2_ NPs were also verified using TEM (Figure 1E). A highly ordered, mesoporous nature of nanoparticles with hexagonal arrays was obviously noticed, which is the major representative property of mesoporous silica nanoparticles. The mesoporous property was also observed after loading the insecticide into FL-SiO_2_ NPs (Figure 1F,H). 

A ZetaSizer Nano ZS Analyzer, based on DLS, was used to measure the diameters of Me@FL-SiO_2_ NPs and FL-SiO_2_ NPs [53]. The average diameters of the two sizes of FL-SiO_2_ NPs and Me@FL-SiO_2_ NPs were (Table 1) much higher than those determined by SEM, possibly due to agglomeration of nanoparticles in solution. Me@FL-SiO_2_ NPs and FL-SiO_2_ NPs exhibited negative charge, showed by the zeta potential of −8.2 and −8.4 mV for FL-SiO_2_ NPs-70 nm and Me@FL-SiO_2_ NPs-70 nm, while −8.2 mV and −8.5 mV were observed for FL-SiO_2_ NPs-150 nm and Me@FL-SiO_2_ NPs-150 nm, correspondingly (Table 1).

TGA was always used to study the decomposition behaviors and thermal stability of chemicals and materials such as methoxyfenozide, FL-SiO_2_ NPs, and Me@FL-SiO_2_ NPs (70 nm and 150 nm) (Figure 2A). The TGA analysis showed that both sizes of FL-SiO_2_ NPs are thermally stable and have constant weight. On the other side, the TGA curves of methoxyfenozide and two sizes of Me@FL-SiO_2_ NPs were altered with the temperature. When the samples were heated up to 500 °C, both sizes of Me@FL-SiO_2_ NPs showed weight loss of about 55%, while a sharp weight loss of approximately 96% was clearly observed in the case of methoxyfenozide. Xu and his coworkers [54] determined that the thermal stability of insecticide such as methoxyfenozide trapped within the FL-SiO_2_ NPs will be increased due to the barrier effect of the nanoparticles towards the volatile products generated by thermal decomposition. This confirmed that FL-SiO_2_ NPs are effectively loaded with methoxyfenozide, exhibitsa strong interaction between methoxyfenozide and FL-SiO_2_ NPs.

The FTIR spectra of methoxyfenozide, FL-SiO_2_ NPs, and Me@FL-SiO_2_ NPs were recorded using a Nicolet 6700 FT-IR spectrometer (Figure 2B). The typical peak at 1065 cm^−1^ in all nanoparticle samples was ascribed to a Si-O-Si (siloxane) stretching vibrational process. Methoxyfenozide displays strong band absorption at 1535 cm^−1^, due to C=O stretch vibrations of the peptide linkages present in methoxyfenozide. The distinctive band absorption of the amide group can be found in Me@FL-SiO_2_ NPs, approving the successful loading of methoxyfenozide into FL-SiO_2_ NPs.

Brunauer–Emmett–Teller (BET) surface area and Barrett–Joyner–Halenda (BJH) pore size were used to determine the mesoporous structure of the two sizes of FL-SiO_2_ NPs and Me@FL-SiO_2_ NPs. The values of the BJH pore diameter (D_BJH_), the total volume (V_t_) and BET specific surface area (S_BET_) are given in Table 1. The type IV isotherm curve (Figure 2C) and the pore size distribution curve (Figure 2D) with an evident step between 0.3 and 0.4 P/P_0_ demonstrate the typical mesoporous structural characteristics of FL-SiO_2_ NPs. Both sizes of methoxyfenozide-loaded silica nanoparticles showed evident declines in pore volume and surface area. The S_BET_ and V_t_ of FL-SiO_2_ NPs (70 nm) decreased from 881 to 121 m^2^/g and 1.5 to 0.5 cm^3^/g, while reduction in S_BET_ (969 to 149 m^2^/g) and V_t_ (1.4 to 0.2 cm^3^/g) was observed in FL-SiO_2_ NPs (150 nm), correspondingly, suggesting that most the mesopores are occupied by the methoxyfenozide molecules.

### 3.2. Controlled Release of Methoxyfenozide

The cargo molecule methoxyfenozide is stable in aqueous solution at specific pH 7.5 value. Moreover, the insecticide methoxyfenozide is generally applied to control insect attacks on several crops. Every organism’s body, including insects has acidic or basic nature [55]. Thus, we chose release media with pH 7.5 value to investigate the release behaviors of the present system. The controlled release profiles, methoxyfenozide (TC), Me@FL-SiO_2_ NPs-70 nm, and Me@FL-SiO_2_ NPs-150 nm at specific 7.5 pH value, were investigated. Figure 3 shows the in vitro release of methoxyfenozide from Me@FL-SiO_2_ NPs at pH 7.5 value. Hollow spheres of FL-SiO_2_ NPs can carry large numbers of cargo molecules in the empty space, which enables the slow and sustained release of insecticide through the mesopores with circular orientation in nanoparticles. At first, the methoxyfenozide was released in a burst, due to existence of methoxyfenozide on and near the exteriors of nanoparticles, which is helpful for immediate control of insects. At optimum pH 7.5, the persistent bioactivity of *P. xylostella* can be obtained through constant release of methoxyfenozide from the mesoporous structure of nanoparticles. The percentage of accumulative release was observed for methoxyfenozide (TC), Me@FL-SiO_2_ NPs 70-nm, and Me@FL-SiO_2_ NPs-150 nm at 7.5 pH values. The accumulative release values reached to approximately 81%, 83%, and 33% for Me@FL-SiO_2_ NPs-70 nm, Me@FL-SiO_2_ NPs-150 nm, and methoxyfenozide (TC), respectively. The enhanced release of insecticide from Me@FL-SiO_2_ NPs at the optimal pH value is due to the instability of Si-O-Si bonds under an alkaline environment, which causes disintegration of nanoparticles. A slow release rate of methoxyfenozide (TC) was observed due to the absence of Si-O-Si bonds, which causes breakdown of silica nanoparticles under optimum environmental conditions. The results predicted that accumulated release of insecticide was enhanced after loading the methoxyfenozide into FL-SiO_2_ NPs, while low release was calculated in the case of methoxyfenozide (TC) due to a lack of FL-SiO_2_ NPs. Recent outcomes are in favor of previous studies which show that sustainable plant protection can be attained through a pH-responsive release pattern [32,54].

### 3.3. Bioactivity Study of Me@FL-SiO_2_ NPs 

The LC_50_ and LC_90_ values were noted as 24 mg/L and 1339 mg/L for methoxyfenozide (TC), while LC_50_ (14 mg/L) and LC_90_ (71 mg/L) were observed for Me@FL-SiO_2_ NPs-70 nm after 72 h exposure, respectively (Table 2). However, LC_90_ and LC_50_ of Me@FL-SiO_2_ NPs-150 nm were 49 and 15 mg/L. After 72 h, the highest mortality, more than 90%, was observed for both sizes of Me@FL-SiO_2_ NPs, while methoxyfenozide TC exhibited 81% at the maximum concentration of 200 mg/L. Similarly, the mortality (%) was higher in insects treated with Me@FL-SiO_2_ NPs with sizes 70 nm and 150 nm than those treated with methoxyfenozide TC. Recent results also showed the significant values *p* = 0.048 and *p* = 0.018 at Me@FL-SiO_2_ NPs-150 nm with 200 and 25 mg/L concentrations, while methoxyfenozide (100 mg/L) exhibited a significant value with *p* = 0.039 (Figure 4).

The recent research predicted that insecticidal activity of methoxyfenozide was increased after loading it into mesoporous silica nanoparticles. Previously, ZnO nanoparticles significantly enhanced the toxicity of methoxyfenozide and ß-cyfluthrin against second-instar larvae of *S. littoralis* [21]. According to Jameel [21], the mortality of fourth-instar larvae of *S. litura* was increased after treating with ZnO NPs with thiamethoxam (10–90 mg/L) than those treated with thiamethoxam. The efficiency of Me@FL-SiO_2_ NPs was enhanced due to better deposition into host leaves and upgraded the penetration power of insecticide into the insect (by physisorption into the cuticular lipids) cuticle. The better insecticidal activity of insecticide-loaded nanoparticles may be due to the nano-size of FL-SiO_2_ NPs, which can hinder the run-off loss of Me@FL-SiO_2_ NPs. The enhanced bioactivity of methoxyfenozide into *P. xylostella* will be checked later through the CLSM study. Our results predicted that Me@FL-SiO_2_ NPs kill the insect more effectively because nano-delivery systems such as FL-SiO_2_ NPs in encapsulation decrease the amount of insecticide used by providing and solubilizing a desirable release of methoxyfenozide through nano-formulations. Moreover, silica-based nano-formulations are designed to enhance diffusion and slow absorption of natural or hydrophobic active compounds, and other purposes that are economically durable and biocompatible [56]. The use of these nano-materials improves the slow release of insecticide rate about 25–75% [56]. The current studies showed that porous hollow silica nanoparticles extended the duration of methoxyfenozide against *P. xylostella* and significantly reduced cytotoxicity; therefore, silica nanoparticles are expected to significantly improve insecticide delivery systems in the future. When we used the silica nanoparticles without insecticides, no mortality was checked. However, thus far, based on our search of all literature, there has been no probing on such detoxification enzymatic studies and control of insects, especially the *P. xylostella* using nanotechnology, and our study is the first report in this area. Generally, the consequences of our research present an encouraging enhancement of toxicity of nano-formulation of insecticide in pest management.

### 3.4. Effect of Me@FL-SiO_2_ NPs (70 nm and 150 nm) and Methoxyfenozide TC on Detoxification Enzyme Activity 

Knowledge of resistance mechanisms in *P. xylostella* against various insecticides is very important in its management [57,58]. The resistance is a function of increased activity of detoxification enzymes, insensitivity of the target site, knockdown resistance, and diminished cuticular penetration [59]. Previous research proposed that insect resistance mechanisms comprise mutations in target places of amino acids for carbamates, organophosphates, and organochlorines [60]. However, resistance in metabolism of insects is the major mechanism in which the production of detoxification enzymes such as acetylcholinesterase, glutathione S-transferases, esterase, and cytochrome P450 monooxygenase become increases [61]. The great expression of P450 for pyrethroids, esterase for novel insecticides, carbamates as well as pyrethroids, and GST for parathion in *P. xylostella* were checked. Xenobiotics including nanoparticles, insecticides, and insecticide-loaded nanoparticles in the organisms can interrupt the normal functioning of enzymes, causing the production of reactive oxygen species, results in reduction or inactivation of detoxification enzymes [62,63]. The relationships between insecticide resistance and enzyme activity have been studied. Development of resistance in insects against various insecticides has been commonly determined by changes in enzymes activities. 

Presently, the effect of two sizes of Me@FL-SiO_2_ NPs (70 nm and 150 nm) and methoxyfenozide TC on the detoxification enzymes (GST, AChE, P450, and CarE) of *P. xylostella* was measured. As shown in Figure 5A, at all concentrations, the GST activity for *P. xylostella* treated with Me@FL-SiO_2_ NPs-70 nm and -150 nm was reduced compared to samples treated with methoxyfenozide TC. The highest GST activity was noted at the highest concentrations, 200 mg/L, of methoxyfenozide, while activity declined in the case of Me@FL-SiO_2_ NPs (70 nm and 150 nm) at the same concentrations. According to Bilal [32], GST played an important role in resistance development in *P. xylostella* against indoxacarb TC, while reduction in activity was observed in the case of indoxacarb-loaded fluorescent silica nanoparticles (IN@FL-SiO_2_ NPs). Similarly, GST activity also declined in treated *D. magna* with various concentrations of CuO NPs and ZnO NPs. Previous studies and our results confirmed that nanoparticles or nano-formulations of insecticides could change enzyme activity [32,64].

For P450 enzyme, its function is to detoxify the xenobiotics and many endogenous substances found in almost all organisms [65,66,67]. Currently, the activity of P450 in *P. xylostella* was at a maximum at the concentration of 200 mg/L post-exposure to methoxyfenozide, while the activity was noticeably diminished by the treatment with Me@FL-SiO_2_ NPs (Figure 5B). There was a prominent increase in activities of P450, esterase, and GST, possibly the major detoxification enzymes responsible for insecticide resistance in *P. xylostella.* The biochemical analysis study predicted that resistance in insects against insecticides was observed by synergistic suppression. Previously, it was reported that silica NPs carrying indoxacarb decreased the P450 enzyme in *P. xylostella* as compared to indoxacarb alone [32].

In the case of CarE, least activity was determined by *P. xylostella* samples after treatment with Me@FL-SiO_2_ NPs-70 nm and -150 nm as compared to methoxyfenozide (TC) at all concentrations. Furthermore, the highest activity was observed at the highest concentration of 200 mg/L after treatment with methoxyfenozide TC (Figure 5C). Similarly, CarE activity was reduced in IN@FL-SiO_2_ NP-treated *P. xylostella* samples as compared to samples treated with indoxacarb alone [32].

AChE is found in all organisms including insects, animals, reptiles, and fish, which degrades the acetylcholine (ACh), a neurotransmitting agent, and is catalyzed into inactive chemicals such as choline and acetic acid [68,69,70]. Recent studies predicted that activity of AChE was also suppressed in *P. xylostella* samples after treatment with different concentrations of Me@FL-SiO_2_ NPs-70 nm and -150 nm, while activity remained high in the case of methoxyfenozide (TC)-treated samples, as shown in Figure 5D.

In addition, several other enzymes such as acid and alkaline phosphatases, also responsible for detoxification of insecticides, ultimately cause resistance in insects [71]. In the current study, two sizes of Me@FL-SiO_2_ NPs showed higher mortality (%) in *P. xylostella*, as well as enzyme activities being suppressed, indicating that nano-formulations with a smart delivery system of pesticide could potentially be used in resistance management strategies against several insecticides.

### 3.5. Confocal Microscopy of Fluorescent Silica Nanoparticles in the P. xylostella 

CLSM was used for imaging of the presence of fluorescent silica nanoparticles in the dissected midgut of treated *P. xylostella* with FL-SiO_2_ NPs. It was also treated with water as control and checked under CLSM to compare with samples treated with FL-SiO_2_ NPs. The treated larvae exhibited a strong fluorescence as compared with the control group, which were without fluorescence under a laser excitation wavelength of 448 nm, as shown in Figure 6. Previously, CLSM showed similar results to confirm the presence of amorphous silica nanoparticles in the midgut of *D. melanogaster*, while no fluorescent signals were exposed from the control group [72]. CLSM was also used to observe the toxicity of SiO_2_ NPs in initial stages of embryo formation in *Xenopus laevis* [73].

Former studies also confirmed the in vivo fluorescence imaging of FITC-CS/γ-PGA and FL-SiO_2_ NPs in the intestine or midgut of insects including worms and *P. xylostella*, which confirmed that nanoparticles could enter the insect body through ingestion [32,74,75]. Moreover, fluorescent double-shelled hollow MSNs were prepared to deliver the fungicide and its presence in the mycelium was proved by visual observation with CLSM [48].

Modern studies prove that nanoparticles could serve as a carrier of the insecticide into the organism, especially the insect gut, which can enter through mouthparts, providing a prospective strategy to overwhelm insecticide resistance.

## 4. Conclusions

This recent study has proved the potential of methoxyfenozide released from FL-SiO_2_ NPs for the control of *P. xylostella*. The Me@FL-SiO_2_ NPs tested here impose a significant negative effect on larval health and ultimately cause death. They exert a range of detrimental impacts on defense mechanisms and insect physiology. The reduction in the activity of enzymes related to oxidative stress and defense is a symptom of cytotoxicity executed by the Me@FL-SiO_2_ NPs. Transmission and scanning electron microscopic evaluation of FL-SiO_2_ NPs and Me@FL-SiO_2_ NPs confirmed the successful loading of insecticide into the NPs. It is noteworthy that Me@FL-SiO_2_ NPs can completely disrupt the enzymatic activities of *P. xylostella* larvae. Moreover, in earlier studies, the insecticides released from silica nanoparticles have already been proven to be safe in controlling any insect models. Therefore, the current study highlights the toxicity of Me@FL-SiO_2_ NPs to *P. xylostella*. Further studies should focus on evaluating the specificity of Me@FL-SiO_2_ NP-mediated toxicity to other important agricultural insect pests. It should also aim to understand of the mode of action of Me@FL-SiO_2_ NP-mediated insect toxicity. These studies will showcase the commercial application of Me@FL-SiO_2_ NPs as a low-cost competent material for use in integrated pest management practices.

## Figures and Tables

**Figure 1 ijms-23-05790-f001:**
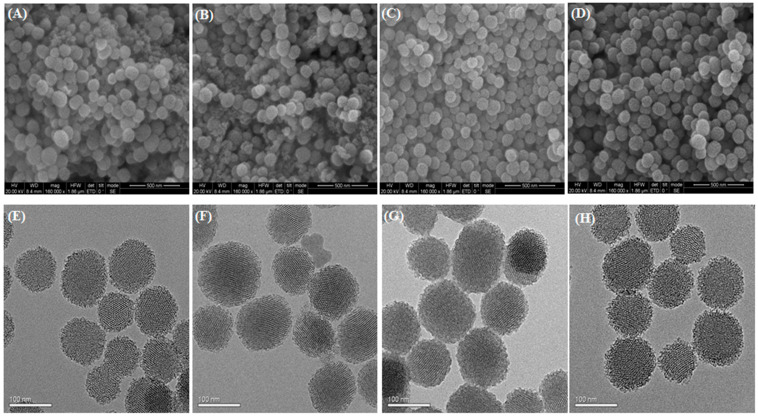
SEM images of FL-SiO_2_ NPs-70 nm (**A**) and Me@FL-SiO_2_ NPs-70 nm (**B**), FL-SiO_2_ NPs-150 nm (**C**), and Me@FL-SiO_2_ NPs-150 nm (**D**); TEM images of FL-SiO_2_ NPs-70 nm (**E**) and Me@FL-SiO_2_ NPs-70 nm (**F**), TEM images of FL-SiO_2_ NPs-150 nm (**G**), and Me@FL-SiO_2_ NPs-150 nm (**H**).

**Figure 2 ijms-23-05790-f002:**
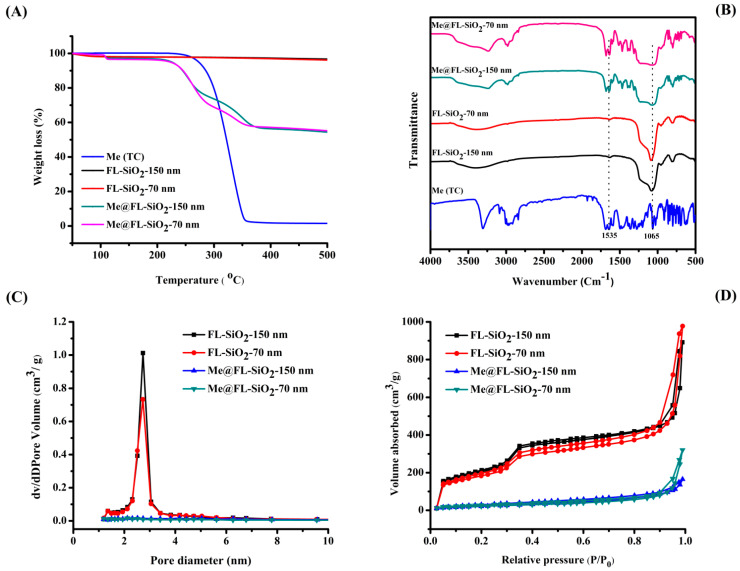
TGA curves (**A**), FT-IR spectra (**B**), Nitrogen adsorption–desorption isotherms (**C**), and pore size distributions (**D**) of methoxyfenozide (Me), FL-SiO_2_ NPs-70 nm, FL-SiO_2_ NPs-150 nm, Me@FL-SiO_2_ NPs-70 nm, and Me@FL-SiO_2_ NPs-150 nm.

**Figure 3 ijms-23-05790-f003:**
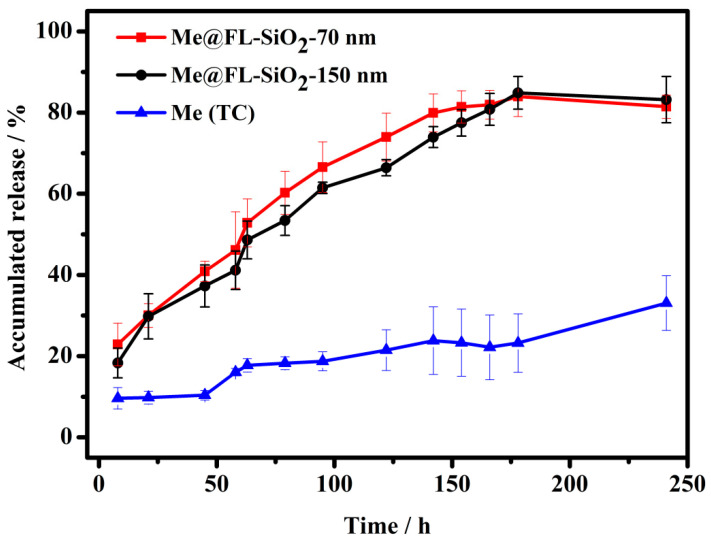
Controlled release profile of methoxyfenozide from Me@FL-SiO_2_ NPs under pH 7.5 value. Error bars correspond to standard errors of three measurements.

**Figure 4 ijms-23-05790-f004:**
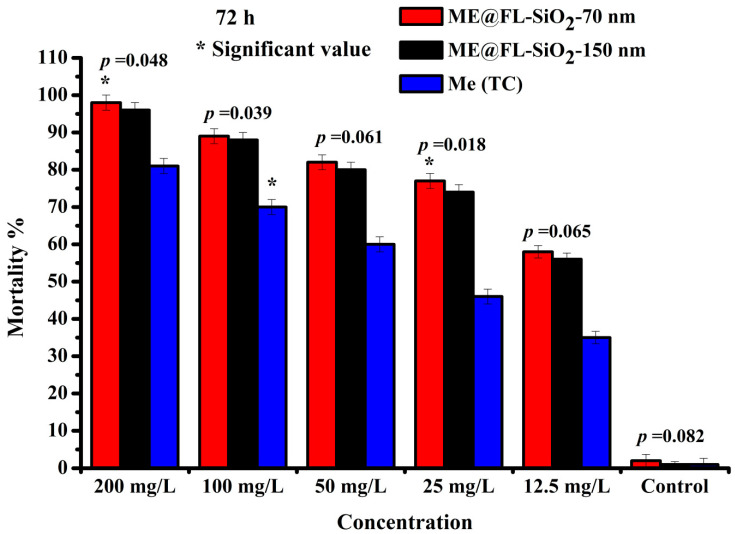
Mortality (%) of *P. xylostella* treated with Me (TC) and Me@FL-SiO_2_ NPs-150 nm and -70 nm under different concentrations. * represents the significant values.

**Figure 5 ijms-23-05790-f005:**
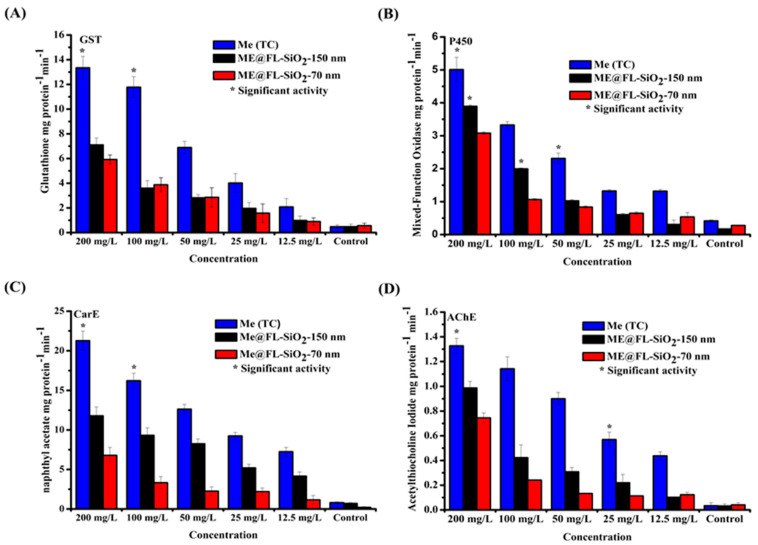
Detoxification enzyme activities in *P. xylostella* larvae treated with Me (TC) and Me@FLSiO_2_ NPs under different concentrations (**A**–**D**). * represents the significant values.

**Figure 6 ijms-23-05790-f006:**
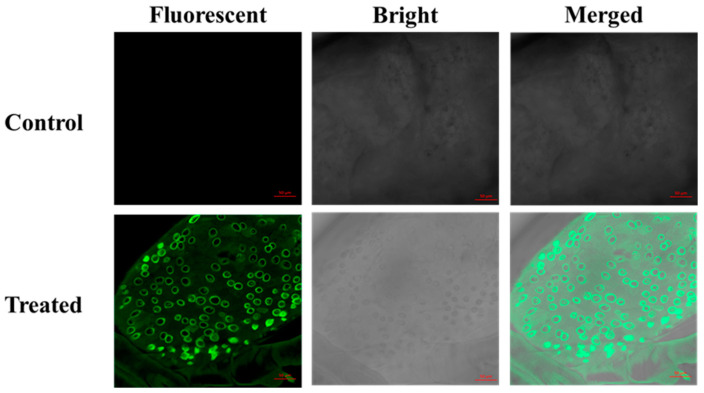
CLSM images of the midgut of *P. xylostella* treated with FL-SiO_2_ NPs (Bar scales: 50 μm).

**Table 1 ijms-23-05790-t001:** Characterization of FL-SiO_2_-70 nm, FL-SiO_2_-150 nm; Me@FL-SiO_2_-70 nm and Me@FL-SiO_2_-150 nm.

Sample	Size (nm) ^a^	Size (nm) ^b^	S_BET_ (m^2^/g)	D_BJH_ (nm)	V_t_ (cm^3^/g)	Zeta (mV)
FL-SiO_2_-70 nm	102	312	881	2.74	1.5	−8.2
FL-SiO_2_-150 nm	168	387	969	2.73	1.4	−8.2
Me@FL-SiO_2_-70 nm	113	534	121	2.25	0.5	−8.4
Me@FL-SiO_2_-150 nm	180	602	149	2.19	0.2	−8.5

^a^ Diameter was estimated by statistical analysis of SEM micrographs of more than 200 nanoparticles. ^b^ Diameter was determined based on dynamic light scattering (DLS). S_BET_: Brunauer–Emmett–Teller surface area; D_BJH_: Barrett–Joyner–Halenda (BJH) pore volume; V_t_: Total pore volume.

**Table 2 ijms-23-05790-t002:** Dose–response toxicity assay of methoxyfenozide (TC), Me@FL-SiO_2_ NPs-70 nm, and Me@FL-SiO_2_ NPs-150 nm to the larvae of *P. xylostella*.

Insecticide/Insecticide@FL-SiO_2_ NPs	LC_90_ mg/L	LC_50_ mg/L	FL ^a^ 95% CL ^a^ mg/L	Slope (±SE ^b^)	Chi ^c^	N
Me@FL-SiO_2_ NPs-70 nm	71	14.4	10.5–18.01	1.024 ± 0.170	1.673	540
Me@FL-SiO_2_ NPs-150 nm	49	15.2	10.8–20.22	1.087 ± 0.155	1.032	540
Methoxyfenozide (TC)	1339	24.74	18.2–31.33	1.393 ± 0.190	0.603	540

^a^ Confidence limits; ^b^ Standard error; ^c^ Chi-square value; N Total number of insects.

## Data Availability

The data presented in this study are available in this article.

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
