# Peer review of "Effects of Methoxyfenozide-Loaded Fluorescent Mesoporous Silica Nanoparticles on Plutella xylostella (L.) (Lepidoptera: Plutellidae) Mortality and Detoxification Enzyme Levels Activities"

_ijms, 2022, doi:10.3390/ijms23105790_

Round 1

Reviewer 1 Report

Overall remarks

Dear Editor

Bilal et al. “Methoxyfenozide loaded-fluorescent mesoporous silica nano-particles for effective control of Plutella xylostella (L.) and their effect on detoxification enzymes activities” titled manuscript evaluates the mortality and detoxification enzyme levels of methoxyfenozide active ingredient combined with silica bonded nanoparticles on diamondback moth. I think that the current manuscript presents compelling results on methoxyfenozide efficiency with the addition of silica nanoparticle. I believe that the manuscript will be welcomed by “International Journal of Molecular Sciences” audience and community and scientific findings will contribute to diamondback moth research. I enjoyed reading and revising the manuscript with its current form. However, I will be more satisfied if the authors would comply with the comments/ changes/ and edits I provided to improve the quality and clarity of the manuscript. For such reasons, I recommend minor revisions to further proceed with publication on IJMS. In addition, I found a couple of grammatical mistakes and words that disrupt the flow of the reading. I suggest authors to check those for clarity and having an additional proof reading.

Please see my edits below.

Title 2-4 I would rephrase the title to reflect your findings better. For example, you are not doing any field work, or you are doing any areawide control. I would not use the word “control”. Your work is more like testing the toxicity and evaluating the mortality. One suggestion would be “Effects of methoxyfenozide loaded-fluorescent mesoporous silica nanoparticles on Plutella xylostella (L.) (Lepidoptera: Plutellidae) mortality and detoxification enzyme levels or activities”.

Introduction

LN 35 and 36 please provide citation.

LN 41 change pesticides to insecticides. The term “Pesticides” is a general term for all compounds including insecticides, fungicides, and herbicides. If you are talking about insects use the term “insecticides”. Please go through your manuscript again and change the wording accordingly. (e.g., LN 432)

LN 44 same as above. If you are talking about insects use insecticides term.

LN 45-47 afford? You mean affect? Please rephrase the sentence for clarification.

LN 48 change word “ruin” to damage or harm.

LN 49 please rephrase the sentence for clarification. For example, “When insecticides are applied to crops, foliage is the first contact parts and then transferred to other plant parts”.

LN 53-56 Have the authors thought about investigating the potential phytotoxicity of the Silica-Methoxyfenozide combination in plants? I understand that they have seen successful results in the lab against P. xylostella, does it mean that the combination of the two compounds applicable in the field?

Please justify.

LN 57-61 Please remove these sentences, it disrupts the flow of your manuscript, and it does not provide any benefit. For example, you are already talking about insecticide application and all other details in the previous paragraph and now here at LN 57 you are going back again and stating the world population and food for people.  I would suggest removing those sentences and write a transition sentence to connect LN 56 to LN 62.

LN 68 NP? Nanoparticles? Add description.

Also, I would be really interested in seeing the structure of your new compound. Where do the Si compounds bind on methoxyfenozide. Do you have structural confirmation of the new compound? Do they bind to -O or -N? Please provide the structure. I am assuming that the Si atoms will bind to terminal -O atoms and form O-Si-O structure.

Source: Pubchem

LN 71 add more details about Methoxyfenozide e.g., mode of action and IRAC group number. Ecdysone receptor agonist, IRAC 18.

LN 73 CRFs?

LN 79-80 please add AChE to the sentence.

M&M

LN 109-116 Do you have a method or step you actually check the concentration loaded into the NP? For example, how do the authors know that concentration loaded into the NP is 30 mg/ml? Could it be less than that? What is the mechanism to check this? Could it be somehow different between 70 nm and 150 nm since there is a difference surface area between the particles?

LN 140 Could you please elaborate how did the authors do sample prep for HPLC analysis? Just putting 10 mg of Me@FL-SiO2 in 10 ml ACN? Is that all? Rule of thumb you have to check your recoveries and extraction efficiency to make sure what you are seeing is equal to what you sampled. Just for the future reference. If you are working on Si particles, do not use Silica bonded columns because Si atoms tend to bond to core silica in the column and you will clog your column after so many injections. For more information read Michel et al. 2012 Journal of Chromatography A, 1245 (2012) 46– 54. However, you can use alternative columns such as PolymerX columns bonded with polystyrene divinylbenzene (PSDVB) from Phenomenex. Polymer structure would eliminate the Si-bonding to the column. In addition, make sure you also use high-density polyethylene tubes and vials for your analysis. Si atoms also bind to glassware in the lab.

LN 167 please specify the variety of Brassica oleracea and write the English name as well- wild cabbage? Also, when you start the sentence with Latin name of the insect or any other organism you do not start as P. xylostella you have to write the whole name such as Plutella xylostella. Please check the whole manuscript to fix the others (e.g., LN 91).

LN 169 please state you also used NP of 70 nm and 150 nm. It seems that you only used methoxyfenozide for this experiment.

LN 174 Do you think 72 h is enough to see methoxyfenozide efficiency 100%? Do you think the compound might need more time to show efficiency since it is a juvenile hormone regulator? Please justify?

LN 188, Please specify the microplates used in the experiments for GST, P450, CarE, AchE. Provide the brand names and how many wells were used for blanks.

LN 201 Please check citation 34. LN 200 says Kristensen but reference 34 is Wang et al. 2020.

LN 208 Please specify those little modifications. If someone tries to replicate the study, they will not know what those modifications are. Each paper should stand on its own.  I would recommend providing clear statements in your M&M section.

LN 228 DTNB? Add 5,5-dithiobis-2-nitrobenzoic acid in parenthesis.

LN 233 Information of statistical analysis is insufficient. Please provide detail explanation for statistical analysis what analysis did you use for which experiments. Please further describe the which tests you used in detail.

Results

LN 303-304 add high resolution images for Fig. 2.

LN 346-349 Move the sentences to M&M, they do not belong to Results section.

LN 355 add a column for LC90`s for all the treatments

LN 358 State whether used SE or SD on the bars of Fig.4.  Also, why did you choose to use bar graph rather than doing sigmoidal curve in for the mortality. This makes it difficult to follow. I suggest changing to figure with sigmoidal curve to provide better understanding on LC50 value visual comparison. If the authors want to keep the Fig. 4 as it is, they should indicate if there is a statistical difference among the treatments either with letters or asterisk on the Figure and providing the p-values. Also, what is the time for the mortality? Add 72 h.

LN 361 and 363 add citation.

LN 405 Fig. 5 please provide info for the error bars in your figures. SE or SD? And provide the statistical outputs if they are statistically significant. This does not tell us anything if they are significant rather than visually seeing the difference in enzyme levels. You can indicate by using letters or asterisk. Also please provide high resolution images for Fig. 5.

LN 405 Fig.5. Could you specify the enzyme units with substrates on y axis? Such as Glutathione conjugate mg protein per min and B-napthol mg protein per min.  I understand that the way you did was another way to describe but I think it would give more information regarding enzymatic reaction.

Conclusion

LN 455 Another comment I have is, how does the combined structure (Methoxyfenozide-SiNP) kill the insect?.Apparently, you see increased mortality (approximately 20% higher with NP). One assumption I have is that NP penetrate more into the insect due to non-polar ability to bind waxy surface of insects (chitin) thus, could cause more or additional cell death (apoptosis). I am confident that if you would have tested the NP only (without methoxyfenozide) you would get some mortality as well. It may not be in methoxyfenozide-TC level, but you would get mortality for sure. I would recommend reading some literature to find the potential toxicity of these compounds and provide information on your conlusion/discussion. Please provide some justification or assumption to your discussion. You provided a sentence at LN 452 saying that they serve as carriers, I assume it should be more than that. I know the fact that some organosilicone compounds are causing death or memory loss on honey bees. Please look at Chris Mullin` s papers on organosilicone and honey bees to understand the mechanism of your compound.

Author Response

Thanks for your nice comments and suggestion. Please see the attachment of responses to your comments.

Reviewer 2 Report

The manuscript entitled "Methoxyfenozide loaded-fluorescent mesoporous silica nanoparticles for effective control of Plutella xylostella L. and their effect on detoxification enzymes activities" is generally well written and the compounds are well characterized and in agreement with the literature.

An observation is tha the scheme 2B is not clear. Additionally,  line 29, Table 3.1??, and Refs 65,66??

Author Response

First of all special thanks for appreciation of our research work. Please see the attachment.
